# A Compact Avalanche-Transistor-Based Pulse Generator for Transcranial Infrared Light Stimulation (TILS) Experiments

Abraham Lopez [1], Haley N. Strong [2], Kendra I. McGlothen [2], Dustin J. Hines [2] and R. Jacob Baker [1,*]

1   Department of Electrical and Computer Engineering, University of Nevada, Las Vegas, NV 89154, USA; abraham.lopez@unlv.edu
2   Department of Psychology, University of Nevada, Las Vegas, NV 89154, USA; haley.strong@unlv.edu (H.N.S.); kendra.mcglothen@unlv.edu (K.I.M.); dustin.hines@unlv.edu (D.J.H.)
*   Correspondence: r.jacob.baker@unlv.edu

**Abstract:** A pulse generator using an avalanche transistor operating in current-mode second breakdown driving a 780 nm laser diode is reported. The laser diode is mounted on the skull of a mouse and used in transcranial infrared light stimulation (TILS) experiments. The output current pulse width is approximately 2 ns in an attempt to generate a true impulse-like optical pulse excitation for the TILS experiments.

**Keywords:** TILS; pulse generator; avalanche transistor

## 1. Pulse Generator Design

A schematic for the pulse generator designed for TILS experiments is seen in Figure 1. An input trigger pulse of nominally 5 V is applied to the left side of the 1N4148 diode through an SMA connector. The 2N3904 is used in current-mode second breakdown [1] to drive the 780 nm laser diode, a RLD78MZA6-00A, which is a common, and low-cost, infrared laser diode used in consumer electronic devices such as printers and optical disk readers. Packaged in this device, in addition to the laser diode, is a photodiode which is not used in this design but is shown in Figure 1.

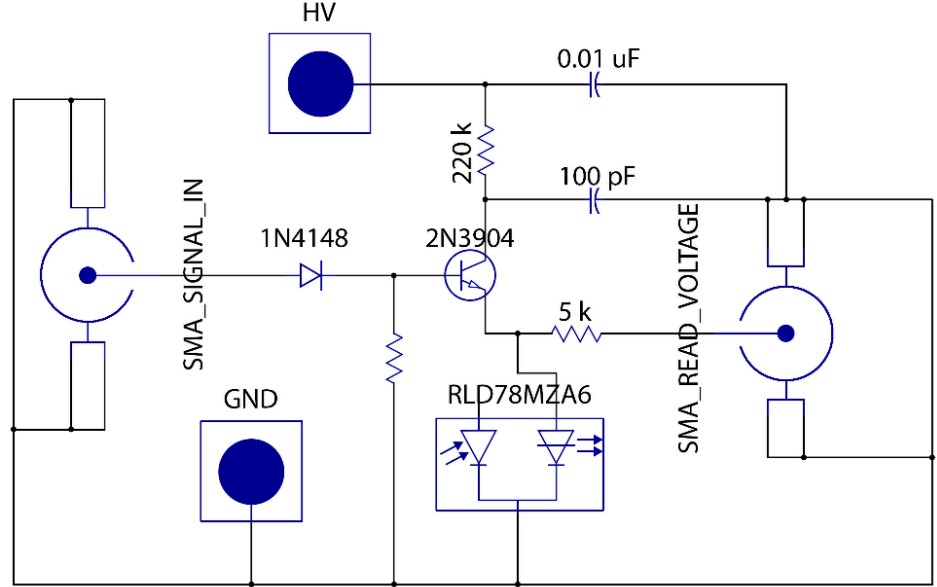

**Figure 1.** Schematic diagram of TILS pulse generator using a standard 2N3904 transistor operating in current-mode second breakdown.

To move the 2N3904 towards avalanche breakdown, so that it can be triggered into second breakdown, a high voltage of around 150 V is applied through the connector labelled HV in Figure 1 through a 220 k resistor to the collector of the 2N3904. This high voltage is decoupled from the supply using a 0.01 µF capacitor. A 100 pF capacitor is used and connected to the collector of the transistor to supply a current pulse to the laser diode. The size of this capacitor is adjusted to set the width of the pulse driving the laser diode. Through experimentation, it was found that 100 pF resulted in a pulse width of roughly 2 ns. Narrow electrical pulse width is desirable to cut off the current to the laser diode prior to the occurrence of the second optical output pulse in the laser diode's relaxation oscillations [1]. When this occurs, the output optical pulse will be less than 1 ns in width. With both the narrow width and very large current through the diode the optical pulse used in the TILS experiments becomes near ideal for impulse response excitation [2]. The output pulse width was monitored with a 100:1 on-board voltage divider implemented with the 5 k resistor seen in Figure 1 and a 50 ohm input of an oscilloscope connected through a cable to the right SMA connector also seen in Figure 1.

## 2. Experimental Methods and Placement of the Laser Diode

### 2.1. Animals and Surgical Procedures

C57Bl6 mice were group-housed under a 12 h light and dark cycle. Food and water were available ad libitum. All procedures were performed following the Institutional Animal Care and Use Committee (IACUC) guidelines at the University of Nevada, Las Vegas. Anesthesia was induced and maintained using isoflurane, and animals were placed in the stereotaxic apparatus. A small incision was made on the scalp exposing the skull, and a small hole was drilled for implantation of the laser diode. Following implantation animals were given a postoperative injection of saline for hydration and singly housed. Mice were given 1 week to recover before experimentation.

### 2.2. Laser Diode Placement and Mounting

One important concern is how to effectively mount the laser diode. Mounting the laser diode on the printed circuit board and then mounting the entire structure directly above, and in, the mouse's head is challenging. The mechanical weight of the printed circuit and diode assembly is difficult to distribute and hold in place. In the work here, the laser diode was connected off-board to the driver through a cable as seen in Figure 2. This way, only the laser diode need be placed directly above the mouse's head, Figure 3. The concern is controlling the shape of the signal so that the laser diode is driven cleanly [3]. As seen in Figure 2, this control was facilitated by using a twisted pair of wires for consistent transmission impedance. It is also desirable to keep these wires as short as possible. The length, however, is limited by the size of the physical devices holding the mouse in place. In the final version of the design, quick disconnects were used, see black connectors in the top image of Figure 4, so that the pulse generator board could be connected to the mouse after the laser diode had been surgically placed in the mouse's head.

### 2.3. Electroencephalogram and Electromyogram Recordings

The recording apparatus consisted of a head mount that connected to a pre-amplifier, commutator, digitizer, and a computer with Sirenia Acquisition (Pinnacle Technologies, Lawrence, KS, USA). The Pinnacle EEG/EMG system is engineered to be artifact resilient, with amplification and filtering at the head by the preamplifier, and further by the data conditioning and acquisition system. Data were acquired with a sampling rate of 1 kHz, low pass 0.1 Hz, and high pass 100 Hz. Mice were placed in a clear plexiglass cylinder with cob bedding and the pre-amplifier was plugged-in to the head mount implant for at least 1 h to habituate to the recording apparatus.

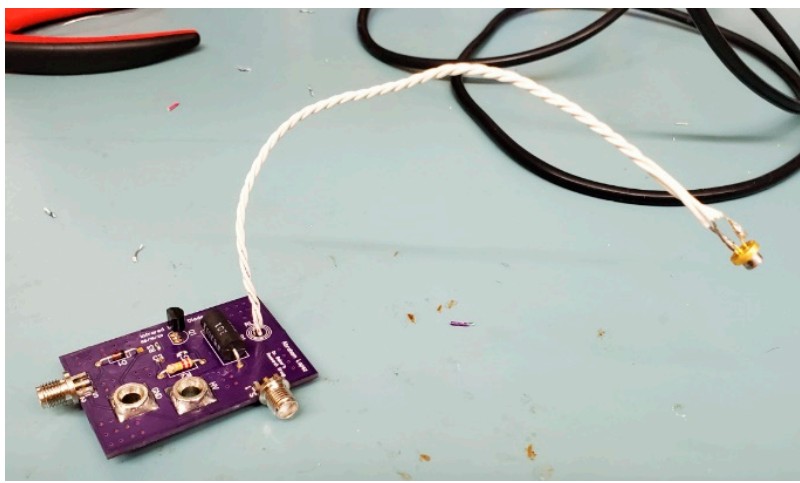

**Figure 2.** Printed circuit board implementation of the design showing the extension of the laser diode.

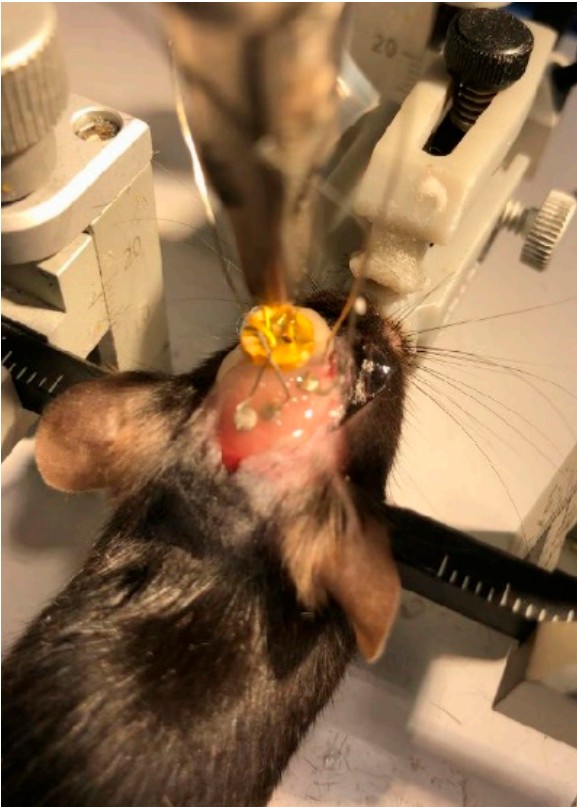

**Figure 3.** Surgically mounting the laser diode in a mouse for TILS experiments.

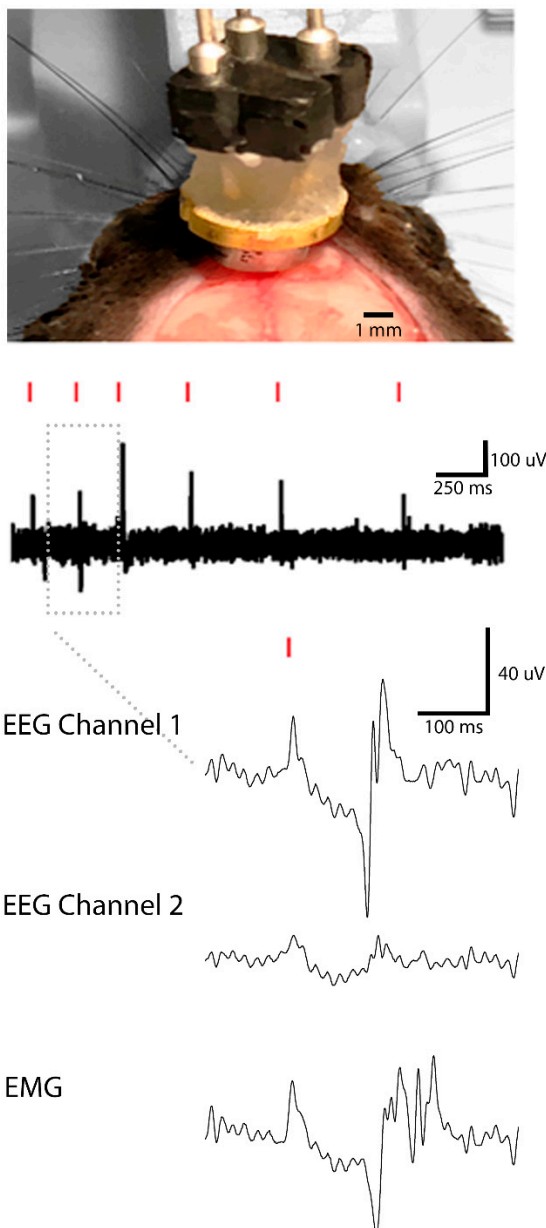

**Figure 4.** Showing a close-up of the head-mounted laser diode after quick-disconnects (in black on top of the laser diode) were added to the diode. Additionally, showing raw electroencephalogram (EEG) trace from channel 1 with multiple TILS stimulations. The red bars indicate when the pulse generator was triggered to generate the stimulus. Below the raw trace are filtered traces of both channels of EEG and electromyogram (EMG) that were simultaneously recorded, expanded from the boxed section of raw trace.

### 3. Experimental Results

Animals were anaesthetized and surgery was performed to implant a head mount with a laser diode, along with EEG and EMG electrodes used to measure electrical changes in the brain and muscles, respectively. A waveform generator is used to trigger the pulse generator in Figure 1. This results in an output optical pulse from the laser diode. Resultant brain activity was monitored using EEG acquisition software from Pinnacle Technologies (Sirenia; Lawrence, KS, USA), Figure 4. Triggering of the circuit resulted in sharp changes in amplitude that could be observed in the EEG and EMG signals. Following stimulation (~100 ms), the EMG shows activity that corresponds with motor movement observable in the body of the animal [4,5]. The change in brain activity induced by TILS is seen in

the zoomed-in trace of the EEG signal as a large depolarization followed by a decay in the tens of milliseconds range. These results indicate the ability of the pulse generator to trigger an infrared signal capable of eliciting changes in electrical activity of the brain, and a corresponding response in the muscle generating movement.

## 4. Concluding Remarks

The small laser diode is capable of delivering fast pulses and is also sufficiently light and compact enough to be implanted. This capability allows for the precise and long-term placement of the laser diode in order to stimulate brain regions of interest. This further allows for longitudinal studies of repeated stimulation and also application in disease models such as photothrombotic stroke and traumatic brain injury.

**Author Contributions:** Conceptualization and design: D.J.H. and R.J.B.; fabrication: A.L. and R.J.B.; experimental plan and methodology: D.J.H. and R.J.B.; electroencephalography implantation and recording: H.N.S., K.I.M. and D.J.H.; writing: R.J.B. All authors have read and agreed to the published version of the manuscript.

**Funding:** This research received no external funding.

**Institutional Review Board Statement:** Animals were cared for according to the NIH Guide for the Care and Use of Laboratory Animals and the protocol for implantation and recoding was approved by the Institutional Animal Care and Use Committee (IACUC) of the University of Nevada Las Vegas.

**Informed Consent Statement:** Not applicable.

**Data Availability Statement:** Not applicable.

**Conflicts of Interest:** The authors declare no conflict of interest.

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
