# Peer review of "A Compact Avalanche-Transistor-Based Pulse Generator for Transcranial Infrared Light Stimulation (TILS) Experiments"

_instruments, doi:10.3390/instruments6030020_

Round 1

Reviewer 1 Report

The manuscript “a compact avalanche-transistor based pulse generator for transcranial infrared light stimulation (TILS) experiments” forms a nice compact piece of work. The manuscript describes an elegant, and a very simple, method to generate nanosecond-scale infrared light impulses for TILS.

I have just a few small things I would like to see further in the work, and thus my recommendation is a minor revision of the work prior to publication.

-

1.

On Figure 4: The large jump-discontinuity in the EEG trace at the time of stimulation looks very similar to say TMS (transcranial magnetic stimulation) artefact on EEG or EMG (especially for monophasic stimulation). Thus, it is hard to say for sure from the shown trace what is a true change in EEG trace due to stimulation and what is an artefact of the current pulse to the LD close to the electrodes (there appears to be a notable drop in high-frequency content lasting a few tens of milliseconds after the pulse, which is potentially real stimulation effect, but this is hard to say for sure without a spectrogram or artefact removal).

Given that the stimulation also affected the EMG (page 3, line 94), it would be more useful to (also) show the EMG trace. It might also be beneficial to try to remove the EEG artefact, although removal of stimulation artefact from EEG is a notoriously hard problem with no easy universal solution.

Alternatively, if the artefact cannot be removed, it might be of use to do some elementary EEG analysis to show changes in the frequency content on specific frequency bands in the pre- and post- stimulus EEG signal.

-

2.

Page 2, line 72. No EEG or EMG system is “artefact free”. This sounds like a term used in an advertisement of an EEG/EMG system. A system might at best “be resilient to artefacts” but not truly “artefact free”. Change the wording.

-

3.

Page 2, line 38–41: Related to the measurement description. If the measurement quality allows easy interpretation, please provide a trace of the 2 ns voltage pulse waveform in the work, to estimate the required light dose.

-

4.

For clarity, on figure 1.

On the component symbols. Add two small arrows pointing into the unused photodiode (PD) of RLD78MZA6 and two small arrows pointing out of the laser diode (LD) of it (potentially also add the second line through the LD body to indicate that it is an LD and not an LED). Ideally, also surround the two diodes with a round or a rectangular blue box to indicate that they are a part of one component. I know that there two features are not included in the datasheet for the component, but they would improve the readability for anyone who sees the figure 1 for the first time and tries to identify where the LD is at.

Remove the small dots from the background of the figure.

Increase the resolution of the figure.

Increase the font size on the pin numbers of each component or, alternatively, remove the (here) unnecessary pin numbers from the components.

-

A few typographical errors:

Page 1, line 10: Change “in the scalp and skull” to “on skull” given that scalp was removed from underneath the diode. (Or, if the “small hole” on page 2, line 49, was drilled through the skull, provide reasoning why the method can be called transcranial if the stimulating light was directly applied to the brain and not applied through the skull (i.e., trans-cranial-ly).

Page 1, line 21: Change “optical disks” to “optical disk readers”.

Page 2, line 29 and elsewhere: Use consistent spacing between values and units, e.g., “150 V” or “100 pF”, “100 Hz”.

Page 2, line 48: Remove the double period.

Page 2 line 63: The sentence “In the final version…” is repeated twice, remove the second copy.

Author Response

  1. Fixed, thank you.
  2. Fixed, thank you.
  3. Good idea, we have added this. Thank you.
  4. We don't have a clearer image to use.

Reviewer 2 Report

The current manuscript was a short communication that describes development and implementation of a pulse generator using an avalanche transistor being used to drive a 780 nm laser diode. Most importantly, the laser diode was able to be successfully mounted into the skull of a mouse during surgery. Accordingly, the laser diode can be used in intervention and basic science experiments to apply transcranial infrared light stimulation to modulate brain activity and ultimately behavior. Finally, the brain activity could be simultaneously measured with EEG from the mouse to quantify changes in brain activity.

I feel this laser diode arrangement represents a practical step forward in the fields of low level laser therapy, light emitting diode therapy, and brain stimulation. The use of light to modulate and enhance muscle and brain function is an emerging topic in the field. Current evidence suggests that applying laser therapy with 980 nm diodes can improve muscle size, strength, and fatiguability as well as have implications for rehabilitations. Furthermore, the use of light applied to various parts of the brain is also currently under study. The use of light in these situations has potential in many fields such as rehabilitation, sports, the workplace, military, and space travel. Thus, this device could allow background information to be developed from animal research to be translated to human applications.

I don’t think the paper has any overt flaws and I have no major concerns. The study described represented some difficult technical challenges and should be published. I have a few minor corrections that the authors should change however, before the paper is published.

Minor comments

  1. Section 2. The first paragraph starting with C57B16 mice…. Seems to have a different format than the other paragraphs. The justification and line spacing look different. Please check this and format the paragraph (or others) according to MDPI formatting.
  2. Line 48 has 2 periods after the word “apparatus”.
  3. I would have liked if there is room for the authors to had a concluding paragraph about the possible applications of the arrangement and why it may be better than current laser diode arrangements for mice. Accordingly, I would have liked to have seen a sentence or two talking about physiological applications of the device in the introduction as well.
  4. Figure 4 looks slightly blurry but it just may be me. The figures are good especially, Figure 3, so perhaps see if 4 can be slightly improved.

Author Response

  1. We thank the author for this comment.  We have added in the text the Sierenia system has a custom head mounted preamplifier that is engineered to be artifact resilient.  The EMG signal corresponds with movements when place in the caudal forelimb area.
  2. We thank the author and have made the change
  3.  We do not have this available from the software.
  4.  Thank you you we have fixed this.

Round 2

Reviewer 1 Report

I do not see the changes described in the response in the submitted version 2 of the manuscript.

Given the last point of this response (which was the point 1 on the original statement), the work does not show that the 2 ns laser pulse stimulated the brain. The shown main feature of the EEG trace is a classic depolarisation artefact of the EEG electrode and the leads, and not a brain response. Thus, I have updated my rating accordingly. Consequently, I will have to reject the revised work.

-

First, page 2 lines 68-70 are still a replication of page 2 lines 65-67. (And all other changes to the work are cosmetic.)

-

Second, there does not appear to be the added trace of the 2 ns pulse waveform despite the response suggesting so with: "3. Good idea, we have added this. Thank you."

-

Third, and main point, the first response item about Figure 4 looking like an electromagnetic artefact has not been addressed. The figure shows raw EEG trace with a stimulus 'response' which looks exactly like a typical EEG artefact and not like a typical EEG evoked response. I suggested either removing the artefact with standard methods or alternatively showing the (more resilient to such artefacts) EMG (which the authors say they had also measured, page 3 line 97). Neither change was done despite the response: "Fixed, thank you."

-

Author Response

We appreciate the time taken by reviewers to carefully assess the work and provide their comments, the feedback provided has improved the work.

To more adequately address the reviewers concerns we have taken the following steps:

We have remade figure 1 to provide higher resolution, improved readability, as well as clearly show the features that we added or adapted in the component. The dots from the background, and the unnecessary pin numbers have also been removed. For enhanced clarity we followed the instructions of the reviewer to 1) add two small arrows pointing in the unused photodiode, 2) add two small arrows pointing out of the laser diode, 3) add a line through the laser diode body to distinguish it from an LED, and 4) enclose the two-diode component in a blue rectangle to indicate that they are a part of a single component.

We have also made improvements to figure 4. In particular we now show filtered traces of both EEG channels recorded, along with the EMG. The EEG traces still show a stimulus artefact (reduced by artefact filter), but more accurately show the alterations in EEG activity that follow because of additional timeline expansion. The addition of the EMG also shows the stimulus artefact, but importantly shows activity that follows by approximately 100 ms that corresponds with animal movement. We have made corresponding changes to the text to help clarify.

We have also made the requested changes to the text, including the suggested revision of the artefact resistance of the EEG/EMG system, as well as fixing the typographical errors noted by the reviewer.

Reviewer 2 Report

The authors have made the changes I recommended last time. These were minor typographical and grammatical errors, which they fixed. In addition, I asked for a conclusion paragraph and this was also added. I think the paper is now ready for publication.

Author Response

We appreciate the time taken by reviewers to carefully assess the work and provide their comments, the feedback provided has improved the work.

Round 3

Reviewer 1 Report

The new Figure 4 and the filtered EMG and EEG traces clearly demonstrate that the short laser pulse did indeed cause suprathreshold stimulation of the brain. With this change in place, the work now demonstrates an elegant brain stimulation method. As such, my recommendation is to accept the work. 

Also, new Fig. 1 is greatly improved, which will help move the field forward. Great work by the authors with this revision!